# Safety of COVID-19 Vaccines in Patients with Autoimmune Diseases, in Patients with Cardiac Issues, and in the Healthy Population

**DOI:** 10.3390/pathogens12020233

**Published:** 2023-02-02

**Authors:** Loredana Frasca, Giuseppe Ocone, Raffaella Palazzo

**Affiliations:** National Center for Drug Research and Evaluation, Istituto Superiore di Sanità, 00199 Rome, Italy

**Keywords:** COVID-19, COVID-19 vaccines, safety, autoimmune diseases, side effects, risk/benefit ratio, myocarditis

## Abstract

The coronavirus disease 2019 (COVID-19) has been a challenge for the whole world since the beginning of 2020, and COVID-19 vaccines were considered crucial for disease eradication. Instead of producing classic vaccines, some companies pointed to develop products that mainly function by inducing, into the host, the production of the antigenic protein of SARS-CoV-2 called Spike, injecting an instruction based on RNA or a DNA sequence. Here, we aim to give an overview of the safety profile and the actual known adverse effects of these products in relationship with their mechanism of action. We discuss the use and safety of these products in at-risk people, especially those with autoimmune diseases or with previously reported myocarditis, but also in the general population. We debate the real necessity of administering these products with unclear long-term effects to at-risk people with autoimmune conditions, as well as to healthy people, at the time of omicron variants. This, considering the existence of therapeutic interventions, much more clearly assessed at present compared to the past, and the relatively lower aggressive nature of the new viral variants.

## 1. Introduction

The pandemic of the coronavirus disease 2019 (COVID-19), which is mediated by the coronavirus SARS-CoV-2, has been a big challenge for the whole world [1,2]. COVID-19 vaccines were considered crucial for disease eradication, and several vaccines have been developed worldwide using innovative or more traditional production approaches. Some of these approaches relied on entire inactivated virus, and these kinds of vaccines have been used mainly in the world’s low- and middle-income countries. As reported by WHO data in 2022, there are several vaccines under various stages of development worldwide, with 153 and 196 vaccines in clinical and preclinical trials, respectively [3,4,5]. The developed products with genetic bases are used mainly in high-income countries (the USA, Europe, Australia), and the use of mRNA-based vaccines is predominant [6,7]. Variability of the SARS-CoV-2 virus is challenging, and the vaccines cannot effectively reduce virus spread, which makes it difficult to achieve herd immunity [8]. Nevertheless, the more “traditional” vaccines and the genetic vaccines seem to have a similar effectiveness. For example, a recent trial on the Soberana vaccine from Cuba demonstrated high immunogenicity, with promotion of neutralizing immunoglobulin G (IgG) and specific T-cell responses against the variants (Omicron variants were not tested as with the genetic vaccines) [3]. Here, we discuss genetic vaccines and, in particular, the most diffuse vaccines in Europe and the USA, mRNA vaccines. Currently, the real effectiveness of mRNA vaccines against Omicron variants is unclear and seems to be lower than that obtained with previous variants, even with a fourth dose [9,10]. Indeed, there are studies showing that, after several months following inoculation, the protection against COVID-19 disease obtained with mRNA vaccines almost completely wanes, unless further doses are taken, and this was noticed already at the time of the spreading of the Delta variant [11,12,13,14,15,16].

Because there are people that have been negatively affected by the COVID-19 vaccinations—as some people have developed conditions including inflammatory cardiomyopathy, such as myocarditis or pericarditis, as well as neurological problems, thrombosis [17,18,19,20,21,22], and other more rare syndromes—it is possible that repeated boosts increase the occurrence of the mentioned adverse events. Given that Omicron variants appear more infectious but less lethal [23,24], the risk/benefit calculation, as underlined by a recent publication [18], may likely require updating. Here, we aim to give an overview of the safety profile of these products and provide molecular details that can explain the risks that are inherent with their repeated administration, on the bases of their mechanism of action. This review takes inspiration from a comment in a recent study published in this journal [25] with regard to the safety, which is distinct from the effectiveness, of these COVID-19 pharmacological interventions in people with autoimmune diseases with a history myocarditis. We take a cue from this topic to discuss the opportunity to administer these products to at-risk people with autoimmune diseases, but also to healthy people, at the time of the Omicron variants [22]. It is important to consider that there have been reports of new diagnoses of autoimmune diseases in temporal relationship with dose administration, although proof of causation is not always clear [26,27,28,29,30,31], whereas several therapies function with regard to COVID-19 disease [32,33]. Most importantly, inflammatory cardiomyopathy (myocarditis/pericarditis) seems to be among the predominant unwanted side effects of the genetic vaccines (see subsequent paragraphs). This is highly relevant for patients with autoimmune diseases for two main reasons. From one aspect, it is well known and supported by a plethora of publications in the scientific literature that autoimmune diseases increase cardiovascular risk [34,35,36,37]. A recent study on a large dataset from patients with 19 different autoimmune diseases in the U.K. identified systemic sclerosis (SSc) and systemic lupus erythematosus (SLE) as some of the conditions mostly associated with cardiomyopathy [34,38]. From another aspect, immune-mediated effects and autoimmunity play a role in cardiac inflammation and myocarditis. Indeed, inflammatory cardiomyopathy is comprised in the group of organ-specific au-toimmune diseases and heart specific antibodies are present in 60% of the affected patients [39,40,41].

A review of the literature about the efficacy of these products is not the object of the present overview, as this topic was widely addressed and reviewed at the time of spreading of the first virus variants, including the Delta variant, and, later on, the first Omicron variants. We debate here the safety aspect, with a final section on the discussion of the mechanisms of escape of mutant viruses, and the ADE phenomenon (antibody-dependent enhancement, see below), which is an additional unwanted side effect of these vaccines. The latter effect, as well as the variability of the virus, which impairs the durability of the protection of COVID-19 vaccines from death or severe disease, is also the object of the present review.

## 2. Safety of COVID-19 Vaccines in People with Autoimmunity and Healthy People

In the following sub-sections, we report the effects of vaccination with genetic vaccines in people with certain autoimmune diseases and in the healthy population, with particular emphasis on heart inflammation.

### 2.1. COVID-19 Vaccination among At-Risk Individuals Such as Patients with Autoimmunity

Autoimmune diseases comprise a group of non-communicable diseases, which affect millions of people in the world; they kill 41 million people each year, which is equivalent to 74% of all deaths globally [42]. Among non-communicable diseases, there are autoimmune diseases. SLE represents the prototype of antibody-driven autoimmune diseases [43]. SLE is an autoimmune disease with multi-organ involvement, and it is characterized by a Type I interferon (IFN-I) and a neutrophilic signature [44,45]. There is not a definitive cure for SLE, and the disease is characterized by alternate remissions and flares.

Other autoimmune diseases, for example multiple sclerosis (MS), are also characterized by flares and remissions. In general, autoimmune diseases are difficult to treat, and the pharmacological treatment includes immune-suppressant and anti-inflammatory therapies, as well as biological therapies directed at different molecules involved in the immune response and immune regulation [46]. The balance between the activation of the immune response to contrast infections and its inhibition to avoid excessive inflammation and disease progression is incredibly delicate. When the COVID-19 immunization campaign started at the end of 2020, more aggressive SARS-CoV-2 variants predominated [47]. This provided the rational for enrolling at-risk patients, including those with autoimmune diseases, to receive COVID-19 vaccinations. These patients were considered to be at a high risk of complications due to both influenza and COVID-19. However, there is an interesting meta-analysis showing that the use of a mono-therapy such as antitumor necrosis factor agents (anti-TNF-α) in these patients was associated with a lower risk of hospitalization and death due to COVID-19 disease [48]. The publications about risk for these patients and other at-risk people for COVID-19 are mostly from 2021 and refer predominantly to the previous SARS-CoV-2 variants. Today, the prevalent variants are derived from Omicron, and all the Omicron variants show so far less lethality [23,24]. Clinical evidence has started to show that symptoms of autoimmune disease could increase after COVID-19 vaccinations. For instance, a meta-analysis in 2021 showed that not only were there appearances of neurological manifestations after the first doses of different COVID-19 vaccines in certain patients, but also more than half of those effects were observed in people with previous history of autoimmunity (53%). In particular, mRNA-based vaccines, followed by viral-vector-based vaccines [49], triggered many MS-like episodes. Among more recent reports, there is a study in MS patients from the U.K. and Germany that reported adverse events after the AstraZeneca and Pfizer vaccines. This study reported a 19% deterioration of MS in the German cohort treated with the mRNA vaccine [50].

Another paper reported a significant increase of relapses in MS patients, especially in females of young age, which also occurred after COVID-19 disease. Even in this study, the data relative to SARS-CoV-2 infections refer to the first waves (from 1 March 2020 to October 2021) [51,52].

A more recent study reports a relapse in 1.31% of the analyzed patients, but 5.5% of the patients reported worsening of the symptoms [53]. New flares have been observed in patients with SLE or Rheumatoid arthritis (RA), as well as cases of new diagnoses of RA after COVID-19 vaccination. We report two examples: Terracina et al. reported a case of a 55-year-old man developing RA flares 12 h after the second dose [54]; Watanabe et al. reported a new onset of RA in a 53-year-old male only four weeks after administration of the vaccine [55]. Again, regarding RA, there have been other reports of flares, although they are considered rare events [56]. There was a study called VACOLUP which included 696 participants and which explored flares in SLE. This study was a cross-sectional and observational study based on a web-based survey between 22 March 2021 and 17 May 2021. In this study, 3% of the 696 patients reported a medically confirmed SLE flare after vaccination [57]. Flares or disease deterioration in 3% to 19% (depending on the study) of patients with autoimmune diseases are not irrelevant.

### 2.2. Risk of Myo/Pericarditis in COVID-19 Infections and COVID-19 Vaccines

Of particular importance are myocarditis and pericarditis, partly because they determine undeniable long-term effects of the adverse event of vaccination. It was not clear immediately after the mass inoculation started that COVID-19 genetic vaccines could be associated with myocarditis/pericarditis and at which frequency. A paper in JAMA [58] reported an incidence of myocarditis cases of 1 in 100,000. For pericarditis, the calculated frequency was 1.8 in 100,000. This means that nearly 3 in 100,000 people, that is, almost 1 in 33,300, could suffer from heart inflammation after inoculation with the COVID-19 vaccine. This paper shows two graphs demonstrating that the risk of both myocarditis and pericarditis increased over time during the COVID-19 vaccine campaign. However, numbers could be higher, as reported in a study on military personnel [59] in the USA, where the incidence of myocarditis is 3.5 times higher in the entire military group analyzed and more than 4 times higher for male personnel, as reported in Table 1 of the study. This translates to a frequency of heart inflammation of about 1:25,000 in male military personnel. The difference between the two studies may be due to the fact that military personnel are subjected to frequent health monitoring, although this is not always guaranteed. An important determinant in these frequency studies is the type of survey (passive versus active), as the data of frequency of heart issues are often derived from a passive survey, which might underestimate adverse events [60]. This is also true for another study, which referred to the database of the “*Clalit Health Services*” in Israel [61]. Despite this limitation, this study estimated a frequency of myocarditis of 2.13 in 100,000, with a much higher frequency (of 1:10,000) for young men (aged 16–29). A frequency of 1:12,361 was calculated in another study in Israel in male adolescents [62]. Another paper also reports an increased risk of myocarditis, especially after the second dose, and particularly after the mRNA-1273 vaccine, with an incidence rate ratio (RRI) of 23.10 (lower with the other vaccines). However, the risk after SARS-CoV-2 positivity was IRR 31.08, but only after 7 days post-positive test [63]; afterwards, IRR tended to decline. Although the paper claimed that myocarditis cases are more frequent in COVID-19 disease than after COVID-19 vaccines, the results of excess presentation of myocarditis cases reported following administration of the mRNA-1273 product are nonetheless high and exceed the frequency of these events after the first seven days following SARS-CoV-2 positivity. In this paper, it is not immediately clear whether the patients with myocarditis were at-risk people, whether they had a mild or a severe COVID-19 disease, or whether they had been previously vaccinated, which may change the meaning of the data. Moreover, the frequency of heart issues was measured over a reduced time period. Indeed, in addition to the problem of the passive survey, the other crucial determinant for studying the adverse events of these products is time, specifically, the interval of observations. Indeed, given the mechanisms of action of these pharmaceutical products and their persistence in the body (see below), heart issues are likely to be observed also later.

For the calculation of the risk/benefit ratio, it is crucial to address whether COVID-19 really constitutes, for example, a major risk of myo/pericarditis as compared to the vaccines. An interesting study is worth mentioning: frequency of myo/pericarditis was examined in a longer follow-up period and in a high number of unvaccinated people in Israel who were recovering from COVID-19 disease [64]. Surprisingly, this study did not detect any increased risk of myo/pericarditis in people that had COVID-19. This is interesting because of the high number of people that were analyzed and the longer follow-up compared to the former studies. These findings seem to contradict data from the CDC (Center of Disease Control, Clifton Road Atlanta, GA, USA), by which authors showed an increment of myo/pericarditis in COVID-19-affected people in hospitals [65]. A frequency of 146 per 100,000 (0.146%) was reported; however, the sample population might not represent the real numbers of COVID-19-affected people at the time but only those hospitalized. Retrospective analyses (such as those performed by the study in Israel [64]) usually rely on a passive surveillance, and one can object that other studies demonstrated a higher frequency of COVID-19-induced myocarditis or pericarditis. Two of these studies [66,67] found that about 20% and 27% of people hospitalized for COVID-19 had myocarditis, even subclinically, because the clinicians measured troponin T in these patients. Such a screening is an example of real active surveillance, although, also in this case, we are dealing with data of frequency in hospitalized patients. To compare the frequency of myocarditis cases in COVID-19 disease to vaccine-induced myocarditis, one should compare studies that are comparable, which means passive surveillance studies in comparison to similar passive surveillance studies, and active surveillance studies in comparison to corresponding studies that also use an active monitoring approach. For instance, there is a study from Thailand [68] which represents a survey undertaken in an active manner and which allowed the discovery of 7 participants out of 300 (2.33%) with at least one elevated cardiac biomarker or a positive laboratory test after vaccination [68]. This study analyzed symptoms, vital signs, ECG, and echocardiography at baseline, day 3, day 7, and day 14 for more than 300 participants aged 13–18 after doses of the COVID-19 vaccine. Cardiac markers were collected systematically. Cardiovascular manifestations, ranging from tachycardia/palpitation to myo/pericarditis, became apparent in 29.24% of patients. Myo/pericarditis was confirmed in one patient after the vaccine. This is important because we have here at least one case of myo/pericarditis for every 300 individuals. Furthermore, 2.3% of cardiac issues occurred in young and healthy subjects, which seems to indicate a higher incidence of heart issues in the vaccinated, much higher than previously mentioned. Moreover, the study also reports two patients with suspected pericarditis and four patients with suspected subclinical myocarditis. The paper declares that the symptoms disappeared in 14 days. A long-term follow-up will be interesting and could inform researchers about the real consequences that these adolescents may have later in their life. Notably, chronic dilated cardiomyopathy (DCM) can be linked to a progressed myocarditis [69].

We may more appropriately compare this study to another report that analyzed, via an active survey, unvaccinated (at the time of the study) young students from several U.S. universities (athletes) [70]. Apparently, the researchers found that 2.3% of these athletes had myocarditis or subclinical myocarditis that was attributable to COVID-19. The risk after COVID-19 and that after COVID-19 vaccination seem thus comparable, according to these data. However, one should consider that the actual risk/benefit assessment of the COVID-19 vaccines is based on the capacity of the initial variants of SARS-CoV-2, up to the Delta variants, to cause myo/pericarditis, as in all the studies cited above. Of interest, almost no data are available on the capacity of the Omicron variants to cause these heart conditions. A paper published in October 2022 reports what is probably one of the very limited examples in which an Omicron variant infection presented with myocarditis in two people [71]. The two patients were previously inoculated with anti-COVID-19 vaccines three times. It is worth noting that the risk of myo/pericarditis after COVID-19 was progressively higher in older patients, whereas for the risk associated with COVID-19 vaccines, the trend is the opposite [65,72].

In general, with the extension of the time frame of observation and the advancement of the COVID-19 vaccine inoculation campaign, the spread of other virus variants, and repetitions of the doses, the majority of infected people are often also vaccinated (before and after disease). Therefore, one should really carefully analyze the data regarding COVID-19 spread and vaccinations to avoid underestimating the effect of COVID-19 vaccines on the development of heart conditions. This is especially crucial in younger patients. In this regard, there is a study that reports higher incidences of calls to emergency departments for heart issues in young people in Israel during the COVID-19 vaccine campaign [73]. In other studies, a frequency of heart inflammation of 1 in 6000 was observed in young people, and even higher frequencies have been reported, as reviewed recently [74,75]. A more recent JAMA paper reported a frequency of 299.5 cases in every 1,000,000 people inoculated in young people aged 18–24 years old (which means 1 case in every 3300 young people receiving the second dose of mRNA-1273 [76]). An Italian study reports that, for young recipients of the vaccine, the excess cases were up to 12.0 per 100,000 [77], whereas a U.S. study reports a frequency of myocarditis of up to 1 in 6250 vaccine recipients [78]. Some of these studies are indicated as active surveys. However, they do not systematically measure any myo/pericarditis marker, which would reveal subclinical myo/pericarditis that may lead to sudden death at a later stage.

One last consideration about the cited papers on vaccine-induced myo/pericarditis is that some of these studies consider only events recorded in hospitals, thus excluding outpatients and underestimating subclinical cases (identified through instrumental/lab tests). Most studies tend to exclude from the count the events occurring in people with previous COVID-19, as the events are attributed to COVID-19. People with previous myo/pericarditis can also be excluded with the assumption that those myocarditis cases are due to individual predisposition and not to the effect of the vaccines [79].

A recent study found a very high risk of myocarditis in young adults, and the authors discuss how booster mandates at universities in the USA are expected to cause net harm in that per each COVID-19 hospitalization prevented, one can forecast at least 18.5 serious adverse events from mRNA vaccines. Among these events, there are booster-associated myo/pericarditis cases in males requiring hospitalization [80]. A recent meta-analysis (not yet peer-reviewed) of papers reporting adverse events declares that many of such papers are not clear. They indicate variable frequency of myocarditis (and of several adverse events other than cardiac issues). Re-calculation by the authors in some cases indicates frequencies ranging from 1 in 5000 to 1 in 200, which should be more carefully analyzed [81]. Regarding incidence of myocarditis in young vaccine recipients, another study, which was conducted in Hong Kong, found that the overall incidence was 18.52 per 100,000 (which is not low: 1.8 per 10,000), with a high incidence after the second dose (21.22 per 100,000). The higher incidence concerns males inoculated with a second dose of mRNA vaccines: 3.7 in 10,000, which means 1 case for every 2700 adolescents with a mean age of 15 years, a population for which the risk for COVID-19 was already low with the previous variants [82]. A recent paper from Canada also reports the frequency of myocarditis cases that required hospitalization. In that study, the frequency went from an overall rate of myocarditis of 0.97 per 100,000 mRNA vaccine doses (not individuals) to an observed rate of 148.32 in 100,000 mRNA vaccine doses after the second dose in males aged 18–29 years who received the mRNA-1273 vaccine. It is worth noting that 148.32 in 100,000 is more than one case in 1000 doses administered [83].

Overall, it seems that the data on myocarditis development after COVID-19 doses are not negligible and are not lower than the cases of myocarditis observed during infections with variants of SARS-CoV-2 that are currently extinct. Given that millions of people have been indiscriminately inoculated, this fact poses some issues. Results from the literature definitely show that myocarditis and pericarditis occur after COVID-19 vaccine doses and are concerning. In addition, by studying molecular characteristics of the myocarditis induced by SARS-CoV-2 (non-Omicron) and by the COVID-19 vaccines, a recent paper found a common pattern that suggests that the two conditions are induced by similar mechanisms [84].

A study conducted by using the system biology approach aimed to shed light on post-vaccine induced myo/pericarditis [85]. This study started from the analysis of the VAERS (Vaccine Adverse Events Reporting System) data in the USA The paper clearly found a signal for myocarditis, especially in males aged 18–29, in as early as 2021. It is interesting for several reasons. The first is that the study also analyzes the effect of other vaccines that use a different technology. The authors show that mRNA vaccines were responsible for 87.19% of myo/pericarditis events reported in the VAERS, whereas the other vaccines with the highest events were the smallpox and anthrax vaccines (based on the use of live viruses), with reported frequencies of adverse events of 12.31% and 3.48%, respectively. The approach of the study identified a signature profile for the interferon-γ pathway in post-vaccine adverse reactions, and this interferon-γ pathway is also increased after viral infection. This may indicate that mRNA vaccines, and possibly the adenoviral vector-based vaccines, act similarly to the live attenuated vaccines. This study also proposes an explanation for the myocarditis observed in young males, as IFN-γ pathways (plus the TNF-α pathway) increase in puberty and later wane, suggesting influences from hormones. A lower sensitivity to the IFN-γ pathway in women may explain the lower incidence of myocarditis cases in females, which are in part attributed to the presence of estradiol in females. IFN-γ is a key component in normal immune responses to viral infections. The data on the triggering of the IFN-γ pathway are also discussed in light of the very well-known effect of this cytokine in increased antigen presentation by endothelial cells, allowing migration of effector T-cells to tissues. These types of studies are useful for addressing the likelihood for certain group of individuals to develop myocarditis with the possibility to re-assess for them the risk/benefit ratio.

Unfortunately, the discrepancies in the data generated by passive and active surveys on vaccine-induced heart inflammation are confusing. We have endeavored to summarize the cited studies and highlight the relative frequencies of myo/pericarditis and other heart abnormalities after inoculation of genetic COVID-19 vaccines and after COVID-19 disease (Table 1). Of note, the frequency of myocarditis highlighted after symptomatic COVID-19 is the one measured at the time of the initial variants; sometimes they included the Delta variants. However, all these virus variants no longer exist, whereas, as mentioned, myocarditis cases reported after infections by Omicron variants have been extremely rare so far. More focused studies and real active surveys are needed, for all classes of ages and in cases of infections with the actual virus variants (or at least the initial Omicron variants).

Lastly, a recent study should still be mentioned for two reasons: frequency of cardiac manifestation and cost of monitoring people after vaccination. This study represents an active although limited survey of young people at school. In this study, after analyzing 4928 students after the second dose of the mRNA vaccine, the authors found that 17.1% of the students were affected with cardiac abnormalities. The affected group transitioned from experiencing palpitations, arrhythmia, bradycardia or altered QT intervals to presenting with myocarditis. Unfortunately, as stated by the authors, not all students could be tested for troponin. The overall incidence of arrhythmia and myocarditis was 0.1%, which means that the most severe manifestations have a frequency of 1 in 1000. The authors mention that the cost of assessing the mRNA-induced adverse events at the cardiac level should stimulate discussion [86].

**Table 1 pathogens-12-00233-t001:** Frequency of myo/pericarditis and/or other cardiac events after COVID-19 and COVID-19 vaccines.

Publication	Active or Passive Survey and Follow-Up Period	Population Analyzed	Frequency of Myocarditis	Frequency of Pericarditis and Other Cardiac Events
Diaz et al. doi: 10.1001/jama.2021.13443 [58]	Passive survey, retrospective cohort study. Time of observation at least 20 days.	Mean age 57 years.	1:100,000 Mean manifestation: 3.5 days.	1.8:100,000 Mean manifestation: 20 days
Witberg et al. doi: 10.1056/NEJMoa2110737 [61]	Passive survey, retrospective cohort study. Time of observation at least 42 days.	Mean age 27 years (and adolescent).	Frequency in adolescent 5.4:100,000. 10.69/100,000 Male (16–29 years).	Not reported.
Patone et al. doi: 10.1038/s41591-021-01630-0 [63]	Passive survey, retrospective cohort study. Time of observation at least 1–7 days.	All ages.	Second dose: mRNA1273/9.8 IRR BNT162B2/1.30 IRR.	Not reported.
Tuvali et al. doi: 10.3390/jcm11082219 [64]	Passive survey, retrospective cohort study. Time of observation at least 10 days.	All population 590.976 cases > 18 years (270,220 M e 320,766 F).	27 cases, high frequency	52 cases with pericarditis.
Buchan et al. doi: 10.1001/jamanetworkopen. 2022.18505 [76]	Passive survey, retrospective cohort study. Time of observation not reported.	All ages.	297 cases, 228/78% M, 24 years. After second dose 207/69.7% M, 24 years. Frequency, M: 299.5 case: 1,000,000 doses.	Not reported.
Mansanguan et al. doi.org/10.3390/tropicalmed7080196 [68]	Active survey. Time of observation at least 14 days.	Only adolescents (13–18 years).	After second dose 12.6:1,000,000 in M.	1 in 300 and 2.3 total cardiac issues
Chua et al. doi: 10.1093/cid/ciab989 [82]	Passive survey, retrospective cohort study. Time of observation at least 20 days.	Only 33 adolescents (12–17 years). Overall incidence 18.52/100,000 doses	Frequency after first dose, M: 5.57:100,000. After second dose, 37:100,000 doses.	After second dose 6 cases, 18.18% pericarditis.
Krug. et al. doi: 10.1111/eci.13759 [78]	Active survey. Time of observation at least 40 days.	253 adolescents 12–17 years (23 F; 230 M) post vaccine: 129 cases after first dose and 124 cases after second dose.	Frequency of myo/pericarditis 93:1,000,000 (M, 12–16 years). Frequency of myo/pericarditis 13/1,000,000 (F, 12–16 years).	208 cases (M and F), high level of troponin with respect to standard.
Massari et al. doi: 10.1371/journal.pmed.1004056 [77]	Passive survey, retrospective cohort study. Time of observation at least 21 days.	Adolescents/young individuals (12–39 years).	441 cases (12–39 years) myo/pericarditis M: 3: 100,000. F: 1:100,000 after first dose, after second dose: 0.7:100,000.	At 53 days after second dose recorded 1 death for pericarditis.
Naved et al. doi: 10.1503/cmaj.220676 [83]	Passive survey, retrospective cohort study. Time of observation at least 7 or 21 days.	Total population >1 year. 99 cases at 7 days (80 M) and 141 cases at 21 days (105 M).	Total frequency at 7 days, M, 12–17 years: 2.64:100,000 and 18–29 years (2.63:100,000). At 21 days in 12–17 years: 2.95:100,000, and 18–29 year 2.97:100,000. Frequency > for second doses (148:100,000).	179 cases myo/pericarditis at 7 days and 308 cases at 21 days. Frequency myo/pericarditis: 1.75:100,000 in 18–29 years.
Montgomery et al. doi: 10.1001/jamacardio.2021.2833 [59]	Passive survey, retrospective cohort study. Time of observation at least 30 days.	Total population 3.5:100,000 for year.	4.36:100,000/M for year.	Not reported.
Mevorach et al. doi: 10.1056/NEJMc2116999 [62]	Passive survey, retrospective cohort study. Time of observation at least 30 days for the second doses.	Only adolescents (12–15 years).	1:12,361 for M, 12–15 years. 1:144,439 for F, 12–15 years.	Not reported
Boehmer et al. doi: 10.15585/mmwr.mm7035e5 [65]	Passive survey, retrospective cohort study. Time of observation not reported.	All ages 146:100,000 for entire population.	187:100,000 for M, and 109:100,000 for F.	Not reported
Shi et al. doi: 10.1001/jamacardio.2020.0950 [66]	Active but hospitalized	Total population. No classification for age and gender, but study on hospitalized patients	19.7% incidence for people hospitalized for COVID-19	Not reported
Guo T. et al. doi: 10.1001/jamacardio.2020.1017 [67]	Active but hospitalized.	Total population. No classification for age and gender, but study conducted on hospitalized patients.	27.80% incidence for people hospitalized for COVID-19.	Not reported.
Daniels et al. doi: 10.1001/jamacardio.2021.2065 [70]	Active survey on COVID-19 students at universities, unvaccinated, with COVID-19 history.	Total population. No classification for age and gender, but study on hospitalized patients.	2.3% incidence for with COVID-19.	High level troponin.
Chiu et al. doi.org/10.1007/s00431-022-04786-0 [86]	Active survey	BNT162b2 vaccine administered to school-aged students (aged 12 to 18 years) through a school-based system.	Abnormal ECG in 1% of the cases.	17.1% of the students had one cardiac symptom after the second vaccine dose.

## 3. COVID-19 Vaccines Safety in Autoimmune Patients and Patients with a History of Myocarditis

On the topics of the risk of myocarditis and the risk of COVID-19 and COVID-19 vaccination for people with autoimmune diseases, the paper by Ramirez et al. [25], which was published recently in this journal, is worth mentioning. Indeed, this paper considered not only the issues in the administration of COVID-19 vaccines to persons with autoimmune diseases, in this case SLE patients, but also their history of myocarditis. Although various papers have focused on the administration of COVID-19 vaccines to people with autoimmunity, this is the first to consider the problem of giving COVID-19 vaccines to SLE patients with the history of myocarditis. In this regard, the paper is interesting because it underlines an important issue to be taken into account when using these pharmacological interventions for at-risk people. In SLE, myocarditis can be present in several patients but not always diagnosed [84,87].

Unfortunately, given the previously reported frequency of myocarditis observed after the use of genetic COVID-19 vaccines, due to the number of patients analyzed in this study, it is unlikely to reveal an effect. The study included only 13 patients, making it difficult to find a case of myocarditis. However, the paper introduces the concept that patients with autoimmune conditions such as SLE can suffer from previous myocarditis, and, therefore, they should be more carefully monitored. This is true also for other autoimmune diseases, for instance, systemic sclerosis (SSc), an antibody-mediated autoimmune condition, which involves the heart. In several patients, myocarditis is present and is difficult to detect without using cardiovascular magnetic resonance (CMR) imaging [88]. It is worth noting that the paper by Ramirez et al. reported that all their monitored patients exhibited an increase of blood-tested troponin T, which is a marker of heart damage [89,90], after the inoculation. This suggests that almost every COVID-19 vaccine injection potentially causes damage to cardiac cells, as troponin measurement always indicates a cardiac injury [91].

Although the marker waned over time, the fact that in a small group of patients, this phenomenon was present in all individuals should call for caution in administering these pharmacological interventions to at-risk people with a history of myocarditis.

The SLE patients were followed for several months, and this is another crucial factor in the study by Ramirez et al., as the risk of myocarditis or pericarditis is high after 14–21 days from dose administration, but subclinical myocarditis can show its effects at a later stage. Without instrumental tests and blood tests, the studies mentioned above would have never discovered a myocarditis, or a subclinical myocarditis. One can hypothesize that heart problems could also become evident after months (for possible reasons for this, see subsequent paragraphs). It is crucial to monitor the patients by paying attention to patients’ reported symptoms but also using instrumental tests, such as echocardiography, and specific blood tests. In the paper by Ramirez et al., this active survey was performed, although not for all patients.

Another important issue in the paper by Ramirez et al. is that more than half of the analyzed patients were taking immunomodulants and immunosuppressants at the time of the COVID-19 vaccine. This may have affected patients’ inflammatory immune responses to the mRNA-based therapy by reducing its amplitude. Thus, it is possible that excess inflammation induced by the vaccine may partly be overcome by the medications that these patients take routinely. This is actually reported in the meta-analysis mentioned above, in which excessive suppressant therapies resulted in more hospitalization and deaths, whereas suppressant mono-therapy was protective [48] in these patients. The scenario of the Ramirez et al. paper is representative of what usually occurs in the rheumatology practice (suppressant therapies are used), which increases the translational value of the data for clinicians. It is likely that taking immunosuppressants can reduce the risk of adverse events in people with SLE and other autoimmune conditions. Of course, whether this translates into a lower effect of protection from COVID-19 disease is not completely clear at the moment. Balancing inflammation with low-dose immunosuppressants could have been a way to minimize adverse events in these patients while still providing protection from severe COVID-19. However, this benefit is not proven. In contrast, it has been reported that autoimmune-disease-affected patients, as well as other categories of at-risk people, such as transplanted patients or patients with cancer [92,93], are likely to develop a lower response to the vaccines. This finding is always taken as a demonstration that these patients should receive continuous boosts. However, considering the additive effect of the doses with respect to continuous Spike protein expression in the body (see below), one should be careful with administering continuous vaccinations. Most importantly, and this fact has relevance for both at-risk and healthy people, these kind of vaccinations have been shown to alter the natural immune responses [94]. A technique of *scRNA-seq* revealed dramatic alterations in gene expression in immune cells after vaccination and a decrease of CD8-positive T-cells. The latter alteration may compromise the capacity of the immune system to combat pathogens with re-activation of endogenous virus, for instance, herpes viruses, especially in immune-depressed patients but also in healthy people [95,96,97,98]. Some of these viruses themselves can cause myocarditis [99]. In this regard, as reported in an exhaustive review about the molecular effects of mRNA vaccines, the type of base substitutions in the liposome-inoculated mRNA [100] could play a role in depressing the normal immune responses. Indeed, mRNA pharmacological interventions for other conditions in which N-methyl pseudouridine (the same base substitution present in the COVID-19 mRNA vaccines) was present have been proposed to suppress or attenuate immunity [100,101,102]. The effect is likely due to the induction of regulatory mechanisms that dampened Type I Interferon production and was highly favored by the type of modified mRNA used. If this modification can be useful in autoimmune settings to dampen excessive immune response to self, the same modification can result in depressed immunity after repeated mRNA-vaccine administrations via similar mechanisms described in the Krienke et al. paper in the journal *Science* [101]. These issues could be studied in more detail [101,102]. Indeed, as discussed below, both the mRNA from the vaccines and the antigen Spike itself, are not transiently (or locally [103]) expressed in the body but persist for relatively long periods of time.

A recent study confirms that compared to healthy donors, SLE patients develop a lower antibody response after COVID-19 vaccine administration, even in the absence of medications that suppress immune responses [104]. The authors claim that auto-reactive T cells had a reduced activation after administration of the COVID-19 vaccination. Among the 36 patients studied, 2 (5.56%) experienced lupus relapses with induction of thrombocytopenia and nephritis, which are not mild conditions. This study somehow confirms that mRNA vaccines can dampen the immune response. Therefore, the general inhibition of autoreactive T-cells is likely due to the general immune suppression that is caused by mRNA vaccines. As mentioned above, the immune suppression can be due to the base substitutions in the mRNA molecule [101].

Lastly, in the paper by Ramirez et al. a significant increase in the constitutional domain of the British Isles Lupus Assessment Group (BILAG) index was observed in SLE patients after COVID-19 vaccine administration. No patients needed any treatment change in the medium term in the therapy. However, the authors agree that a regular monitoring of patients with autoimmune diseases, especially in case of more severe phenotypes, should be part of their standard care. In consideration of the elevation of heart damage markers, the increase of BILAG, and the indication in the literature that myocarditis frequency induced by COVID-19 is not more frequent and riskier than the myocarditis induced by the vaccines, the risk/benefit ratio of continuous dose administration may need revision. This revision is especially needed in the case of young patients, both in the at-risk and in the healthy population. Not least of all, SLE patients can often develop renal issues (lupus nephritis), and a recent study found a doubled risk of disease relapse in patients with a renal disease, although it considered the COVID-19 vaccination safe for these patients [105].

## 4. Possible Mechanisms of COVID-19 mRNA Vaccine-Induced Tissue/Organ Damage and Virus Immune Evasion Strategies

In this section, we describe the molecular mechanisms that can explain the genetic COVID-19 adverse events, as well as the immunological mechanisms at the basis of the escape of variant viruses from immune responses.

### 4.1. Spreading and Persistence of the SARS-CoV-2 Spike Protein in the Body

At the beginning of the COVID-19 immunization campaign, many mass media and organs of health services all over the world repeated that the inoculated material would remain in the deltoid muscle, and only for a few days. The perception by the public was that the mRNA is quickly degraded, which does not apply to the modified mRNA used in the COVID-19 vaccines [100,103,106]. Bio-distribution studies, such as in ref. [103], on liposome micro-particles (LNPs) showed that the material does not stop at the inoculation site. In a later study, the authors propose a new type of mRNA vaccines that use a different type of lipid micro-particles to encapsulate the mRNA. Indeed, the authors declare that this is useful “*to allow the retention of the vaccine particles at the injections site, thus preventing vaccine particles from triggering organ-specific side effects*” [106]. These findings are relevant at least for mRNA-based products. However, DNA vaccines can have similar effects, especially in the case of uncontrolled Spike translation, and high licking from tissue [106,107,108]. At present, several papers in the literature demonstrate that the mRNA vaccines and the translated Spike travel to various body districts, with an expression that it is not so transient [106,107,108], a concept that is reviewed also in [109]. The mRNA product Spike protein persists in lymph nodes for at least two months and is present in micro-vesicles for at least 3 months after inoculation [106,107,108]. Spike, especially its subunit 1 (S1), circulates in blood after inoculation for up to 29 days, as shown in another study [108]. In people with no apparent adverse effects during the short time period of observation following inoculation, a mean of 50/70 pg/mL of Spike protein was measurable in blood [108]. Interestingly, this concentration is nevertheless in the same range of the quantity of Spike measured by the same authors in another study, in which the presence of Spike (S1 subunit) in the circulation of people hospitalized for COVID-19 was detectable [110]. In that paper, the criterion chosen by the authors to categorize “low” and “high Spike patients” was set at 50 pg/mL (so this concentration was considered relevant). The circulating highest S1 levels were the levels correlating with a severe COVID-19 case. This may reflect a higher viral load in these severely affected patients. It is also possible that the association of a higher concentration of Spike protein (and especially the S1) with COVID-19 severity can also reflect the intrinsic toxicity of the Spike protein itself (see paragraph below).

In a recently published study which was directed by the same principal investigator of the two papers mentioned above and which analyzed myocarditis cases in adolescents, the authors documented higher expression levels of circulating long-lasting Spike proteins in patients with myocarditis as compared to patients without myocarditis [111].

It is interesting, in this regard, that levels of Spike protein in a woman with adverse events after inoculation were much higher in circulation [112]. Notably, the Spike protein has been found in particular types of macrophages after 16 months from the last inoculations [113]. Interestingly, recruited monocyte/macrophages play a role in heart inflammation, and a transcriptomic analysis after mRNA vaccination revealed a profound alteration of these cells in people with vaccine-induced myocarditis [114]. If the recruited monocytes/macrophages express Spike, and this process is not excluded from the work in ref [113], the resolution of any inflammation could be delayed. Thus, expression of Spike by heart-infiltrating macrophages is worth assessment in future studies.

The protein Spike was also visualized in cardiac biopsies from people with myocarditis after COVID-19 vaccine inoculation, who exhibited a consistent infiltration of immune cells into their hearts [115]. Spike, or the mRNA encoding Spike, could have traveled to the heart, provoking the unwanted effect of activating a cytotoxic response against this organ. It is worth noting that this phenomenon was observed with different types of vaccines, both RNA and DNA COVID-19 vaccines. Spike was recently visualized in the heart and brain of a person who died 15 days after the third dose of an mRNA vaccine [116]. Spike was detected in herpes zoster skin lesions of an inoculated person that suffered from this infection after the inoculation [117]. The mRNA encoding for the Spike protein was detected by in situ hybridization in a liver biopsy from a patient who presented with hepatitis twelve days after the Pfizer vaccine [118]. Interestingly, a previous paper analyzed the cell infiltrate of a liver biopsy from a patient suffering with hepatitis after COVID-19 vaccination, and the biopsy was shown to contain activated Spike-specific CD8 T-cells, which were identified by peptide-MHC-tetramers [119].

The two examples in the paper by Martin-Navarro et al. and Boettler et al. [118,119], demonstrate what was already discussed and illustrated in a previous paper [120], which underlined how “*every human cell that intakes the LNPs and translates the viral protein (in case of the mRNA vaccines), or that gets infected by the adenovirus and expresses and translates the viral protein (in case of the adenovirus-based vaccines), is inevitably recognized as a threat by the immune system and killed*”. Thus, the immune response will always start as a cytotoxic insult in this case. If the antigen is expressed in the wrong place (in this case, the liver), inflammation will occur (hepatitis). Indeed, the antigen Spike is not only taken up by the cells but is also produced endogenously due to the genetic materials internalized. This also implies that its degradation will also proceed via the proteasomal pathway, leading to massive (in case of high translation) cross-presentation via the MHC I protein complex, which is located on the cell membrane of potentially any types of nucleated cells, driving the cytotoxic effect of the CD8 lymphocytes. Usually, the cross-presentation pathways occur in specialized antigen-presenting cells of a particular type called dendritic cells, which are the ones that take the antigen to the lymphatics to prime the adaptive immune cells [121]. Genetic vaccines, especially the mRNA vaccines, may therefore behave similarly to viruses without a specific cell tropism [122], thus altering the normal interplay between the immune system and the pathogens. Here, the antigen can enter, be expressed for a long period of time, and drive cross-presentation in any kind of cells among the immune cell pool. Any immune cell will be perceived from the adaptive immune system as being infected and will be destroyed, potentially inducing immune suppression. This is why this paper called for an in-depth biodistribution evaluation for both the mRNA and the DNA vaccines [120]. Indeed, the author recalls a pharmacokinetic study performed by Pfizer for the Japanese regulatory agency in which LNPs were found to accumulate in the spleen, liver, pituitary gland, thyroid, ovaries, and in other tissues.

All these papers concur to sustain the results of the recent and past studies, which show that a liposome has the capacity to travel to various body districts [103,123]. Unfortunately, the same may happen with DNA-based vectors [115]. They also definitely indicate that the expression of Spike after inoculation is not transient but can last many weeks or months. This evidence raises the question as to whether it is correct to consider any adverse events of the COVID-19 vaccination exclusively within 14–21 days after inoculation, given that the inoculated products persist for longer. Cosentino M. et al. discuss that mRNA vaccines must be considered to be pharmaceuticals, and their pharmacokinetics should be studied in greater detail [124]. Both mRNA and Spike have been found in the breast milk of vaccinated women, which demonstrates that these products travel into the body and can be excreted with biological fluids [125].

As mentioned above, the IFN-γ pathway induction has been proposed as an important component in the induction of mRNA vaccine side effects [85]; the authors propose the concept that mRNA vaccines act similarly to live-attenuated vaccines. The same IFN-γ signature has been found in a subsequent study [126], which also used system biology and transcriptional signature analyses, and could also explain the mechanism of thrombosis (which is also related to heart issues). One of the most relevant proteins up-regulated by IFN-γ is IP10 (interferon gamma-inducible protein 10), which is key in thrombosis and cytokine storms. The mRNA vaccine BNT162b2 was found to give a signal similar to the LPS-induced activation of platelets, which release, among several factors, PF4 (platelets factor 4), which is also known as CXCL4. We would like to note that the pathways highlighted by these studies are highly relevant for the pathogenesis of autoimmune diseases. Both IP10 and CXCL4 are elevated in vasculitis, and both CXCL4 and IP10 are known to be up-regulated and play a significant role in various chronic diseases, among which are SSc, SLE, and psoriasis [127,128,129,130].

### 4.2. Pathogenic Role of the Spike Protein of SARS-CoV-2

The bio-distribution of mRNA and Spike, the relatively long persistence of this protein in inoculated people, and the presence of the protein in the district of tissue damage following the adverse events reported above prompt questions about the role of the Spike protein produced after vaccine inoculation. Does this Spike interfere with the natural physiology of the vaccinated person, contributing to tissue/organ damage and, ultimately, in the worse scenario, to death? Indeed, one should consider that the Spike antigen (and the modified mRNA itself) is not a biologically inactive factor but can enter into a number of molecular pathways occurring in an organism, including pathways driven by anti-oncogenes [102]. The administration to animals of the sole Spike protein recapitulated the majority of the features of the first COVID-19 disease, suggesting that Spike exerts a consistent part of the toxic effects of SARS-CoV-2 [131]. The effect of Spike of SARS-CoV-2 has been studied in vivo in animal models and in vitro on immune cells and endothelial cells, and there is a plethora of papers on this topic. Spike can damage cardiomyocytes [132] and cardiac pericytes [133], and has a series of pathogenic effects, including interference with pathways at work to keep cancer development in check (for review, see [102]). Spike also independently causes cardiovascular disease [134]. Intravenous injection of COVID-19 mRNA from vaccines induced myo/pericarditis in mice [135]. This paper may indicate that the Spike protein encoded by mRNA vaccines also possesses a pathogenic effect (it is not different in function from the natural Spike). More side-by-side studies using natural and vaccine-produced Spike are needed. This implies that high levels of circulating Spike protein may be harmful. The obvious question is whether the occurrence of adverse events is somehow related to the amounts of toxic protein expressed. Spike may reach vital target organs via the circulation. Some people could produce more Spike or produce it in the wrong place. Indeed, liposomes enter any cells and cannot distinguish among tissues. Liposomes can also enter and induce Spike expression in immune cells. Indeed, the mRNA vaccine was shown to reprogram both adaptive and innate immunity [136], thus interfering with natural immune responses. The changes in immunity may be transmitted to the next generations in animal models [137]. Spike, by binding to its receptor ACE2, can change the catalytic activity of this receptor and enzyme or directly down-regulate the receptor, impeding its functions [138]. ACE2 is important in dampening inflammation and blood pressure, and an increase of blood pressure that lasts for a few days has been observed after COVID-19 vaccine administration [139,140]. In one case, 1 out of 797 participants was hospitalized due to high blood pressure after COVID-19 vaccine administration, according to another study [141]. Although considered rare, an increment of blood pressure, even transient, in at-risk people with heart conditions or with stable high pressure can be fatal. Spike damages endothelial cells in animals, promotes inflammation and cell apoptosis, and disrupts blood–brain barrier integrity [142,143,144,145]. Spike induces endothelial inflammation mediated by integrin signaling [146] and impairs endothelial cell functions via ACE2 [147]. Spike persistence and activity may be responsible for the manifestation of long COVID-19 [148]. This antigenic protein can also activate the complement cascade by inducing platelet aggregation [149], which may account for thrombosis induction, a dangerous adverse reaction caused by these vaccines, as reported above. Spike mediates damage of hematopoietic stem cells [150] by activating the inflammasome. It changes the metabolism of brain endothelial cells and destabilizes microvascular homeostasis [151,152]. It is worth noting that Spike sequence presents an amino acid fragment with a “superantigen” character [153], which may favor inflammation and cytokine storm [154]. Superantigens [153] are a group of molecules that have in common an extremely potent stimulatory activity for T lymphocytes. The prototype superantigen is the staphylococcal enterotoxin B (SEB), which is produced by Staphylococcus aureus and Streptococcus pyogenes. Structural similarities between SEB and a SARS-CoV-2 Spike protein fragment have been described [154]. This superantigen effect of Spike could explain “*multi-system inflammatory syndrome*” (MIS-C) in children/adolescents after COVID-19, a phenomenon observed also after COVID-19 vaccines [155,156,157,158,159]. However, a recent paper demonstrated that Spike is not able to act as a superantigen in human cell lines in vitro [160]. It will be interesting to ascertain the effect of smaller parts of the Spike protein in regard to their possible cytotoxic effect to understand what is causing such high inflammation.

Spike is also responsible for syncytium formation that mediates lymphocyte elimination [161], an effect not shared by Omicron [24]; it concurs with oxidative stress by inducing macrophage apoptosis [162]. In conclusion, there are a myriad of reports about the pathogenic effects of Spike of SARS-CoV-2 (regarding the Spike of the initial variants) in the current literature. One preprint publication on Spike demonstrated that the protein enters the nucleus in human epithelial cells due to presence of a novel nuclear localization signal [163], which is absent in other coronaviruses. Spike could shuttle mRNA into the nucleus, a phenomenon that could have several implications for the genetic maintenance of cells [164].

### 4.3. Mechanism of Immune-Evasion of Mutating Viruses and Vaccines

Another issue in the production of genetic vaccines, but also in more traditional vaccines based on the use of Spike as a unique antigen, is the fact that RNA viruses are usually prone to mutate [165]. Among these viruses, human immunodeficiency virus (HIV) and hepatitis-C virus (HCV) are the most variable, and this variability made the development of vaccines a challenge [166,167,168,169]. Flu vaccines do not always work properly [168] due to similar mechanisms (see below). Indeed, a pitfall is due to the formation of viral escape variants and to antibody-dependent enhancement, which also occurs in COVID-19 (ADE) [170,171,172,173]. ADE is a phenomenon by which the anti-virus antibodies do not neutralize variant epitopes but instead help the mutant virus to enter into the cells, paradoxically increasing infectious potential. It can be linked to the well-known phenomenon previously named the “*original antigenic sin*” phenomenon, which is also called “*immune-imprinting*”; this phenomenon is imparted by the recognition of previous viral variants [170,171,172,173]. Immune imprinting occurs when the immune system has recognized a certain virus variant in the first place, then later encounters a second, very similar variant. The phenomenon of immune imprinting, which spoils the immune defense mechanism and causes virus escape, has been known for decades [170,171,172,173,174]. It concerns antibodies but also responses of T-cells. Both cytotoxic and T-helper cells can be improperly activated in the presence of virus variant epitopes [174,175]. T-cells are crucial in immunity and in vaccine induced immunity, as they orchestrate cytotoxic T-cell activation and the humoral immune responses (follicular T helper cells, Thf, are necessary for the establishment of neutralizing antibody response), and this is true for the development of the anti-SARS-CoV2 immune response [176]. However, spontaneous mutations at T-cells receptor (TCR) contact sites within individual viral epitopes can, in certain circumstances, abrogate or “antagonize” the recognition of the corresponding wild-type epitope; such mutations may contribute to viral persistence. Seminal papers in the past have reported the phenomenon of TCR antagonism: T-cells that are specific for an antigenic epitope are unable to respond, or they respond in an altered way, to a second antigenic epitope, which we define as an altered peptide ligand (APL) and which is very similar to the antigenic epitope encountered in the first place [174,177]. The APL effect was demonstrated for the hemagglutinin of influenza (HA) antigen [177] and later for the recognition of HCV variable epitopes [178]. HIV variants were shown to act as partial agonists, that is, partial activators of the TCR [179]. Given that the antigen Spike of the actual mRNA vaccines, even the new ones, is derived from coronavirus variants that are no longer predominant, the phenomenon of TCR antagonism and immune imprinting may be at work. At one side, given the persistence of the Spike from the vaccine reported above, it is likely that epitopes derived from the new variants will be presented to the adaptive immune cells, together with the Spike encoded by mRNA products. Interaction of a TCR with an APL can result in dramatically different phenotypes of the T cells, ranging from induction of selective stimulatory functions to a complete switch-off of the T-cell functional capacity [179]. A vaccine against several proteins, or a vaccine directed at a less variable region, could be more effective and may attenuate these mechanisms of escape acting on T-cells. In turn, as T-cells are necessary for the production of neutralizing antibodies, an inefficient neutralization of the new variants may occur [180,181]. These pathways, together with the incapacity of the antibodies to neutralize the new variants, can be at the base of ADE. Updated vaccines may not overcome this mechanism because new variants are continuously spreading all over the world.

### 4.4. Autoimmunity after COVID-19 Administration

As already mentioned, there is clinical evidence of autoimmunity and autoimmune disease onset occurring after both SARS-CoV-2 infection and vaccination with COVID-19 genetic products [182,183]. Interestingly, the Spike-binding receptor ACE2 becomes the target of autoantibodies [184] in COVID-19. It will be of importance to test whether vaccines induce these kinds of autoantibodies. It is worth noting that anti-ACE2 antibodies are already present in patients with vasculitis and SLE as part of these patients’ autoreactive repertoires [185].

There is some in silico evidence of potential cross-reactivity between the Spike protein of SARS-CoV-2 and human self-proteins [186,187]. In keeping with this phenomenon, the monoclonal human antibodies to SARS-CoV-2 react to multiple autoantigens including heart antigens in vitro [188]. There are reports of histopathological evidence of myocardial inflammation in subjects with post-vaccine myocarditis with lymphocyte infiltrate, which is suggestive of the presence of autoimmune-like attack [189,190].

The list of modes of possible “molecular mimicry” is consistent in the literature, and we cannot cite all papers here. There are also reports that denied evidence of cross-reactivity between the Spike protein sequence and classical myocarditis-associated auto-epitopes [191]. Of course, the development of autoimmunity after COVID-19 vaccination may be due to a particular predisposition of the individual person. This is the reason why each individual receiving one of the COVID-19 vaccines currently in use needs an anamnesis before taking further doses. Indiscriminate mass-vaccination is not the strategy, especially at the present stage, which is characterized by a lower lethality of the new variants and an established protocol for cure. An autoimmune-like attack can occur if the genic information for Spike is transported to a specific body district, favoring Spike expression in unwanted tissues (for instance, vital organs such as the liver or the heart) and Spike epitope presentation to T-cells. Consequence of the mechanism of action of these vaccines could be an autoimmune-like attack by T-cells to the organ, as if the virus [115] infected the latter. Suggestions about the role these mechanisms play in organ inflammation have been reported above for post-vaccine-induced hepatitis cases. Indeed, not only myocarditis but also hepatitis cases have been observed after vaccination [192,193,194]. It would be preferable for mRNA- and even DNA-based vaccines to induce a local reaction (as the classical vaccines), instead of a systemic-like reaction, which mimics a disseminated infection.

Going back to the paper by Ramirez et al. [25], no clinical evidence of myocarditis exacerbation was found in the SLE patients analyzed. This is a good start, but, again, wider studies are needed. Exacerbation of autoimmunity can develop later. Apart from the mechanism described regarding Spike expression in unwanted locations of vital organs, autoimmunity that is already established can increase due to continuous dose administrations. The authors tend to exclude *de novo* induction of autoimmunity in SLE patients, but this needs demonstration. Indeed, it has been observed that patients with an autoimmune disease can suffer from additional autoimmune conditions [195] and are susceptible to develop a more severe and systemic disease at certain points of their life. For instance, patients with psoriasis can develop psoriatic arthritis in up to 30–40% of the cases [196], and people with cutaneous lupus can develop SLE in up to 18% of cases [197]. The paper by Ramirez et al. reports that adverse events can be higher with natural infections than after vaccination, both in the general population and in patients with immune-mediated disorders. We discussed above that the scenario reported by Ramirez et al. is not really the case, at least for myocarditis. Most importantly, one should always be aware that this assumption is not confirmed with the Omicron variants. Unfortunately, depending on the type of mRNA vaccine, adverse events can be important, especially if the heart is damaged. The adverse events comprise a plethora of different manifestations, each of which is rare on its own, but these manifestations are not rare anymore if considered as a whole. If the vaccines prevented infections, continuous inoculation could make sense. Because the people who received three or more boosts can still have symptomatic infections and risk hospitalization, the risk could be twofold: the risks of COVID-19 disease and COVID-19 vaccination may act in an additive manner.

## 5. Conclusions

This overview on COVID-19 and COVID-19 vaccine adverse events does not aim to discuss the effectiveness of COVID-19 vaccines against the original and the early SARS-CoV-2 variants, as that effectiveness was documented by publications at the first launch of the genetic vaccines. Seminal publications showed protection from death and severe disease after two months from vaccine administration. Several studies have documented a quick drop in the efficacy of these substances, a drop which is more evident after the diffusion of the diverse Omicron variants. Because many studies indicate that the actual virus variants are less lethal, and that effective therapies to treat COVID-19 disease exist, this may be the right time to revise the risk/benefit ratio of these pharmacological interventions. Now an additional factor, which was lacking at the time of the first efficacy studies, is that a great number of people are naturally acquiring immunity also through infections, including pauci-symptomatic infections. Thus, at present, it may be possible and useful to reflect on the documented adverse events of these gene-based vaccines. A small study, after analyzing the U.K. Health Security Agency data, revealed that the mortality rate in unvaccinated people (for non-COVID-19 causes) was lower than that observed in the people who had received at least one COVID-19 vaccine dose [198]. A recent document from the “Office for National statistics” in the U.K. (https://www.ons.gov.uk/peoplepopulationandcommunity/birthsdeathsandmariages/deaths/datasets/deathsbyvaccinationstatusengland) (accessed on 10 October 2022) reports data of mortality for COVID-19 and for all causes excluding COVID-19 at the time of the COVID-19 vaccine campaign. An accurate and transparent statistical analysis of such data, which should take into account all the variables involved, can clarify the real effects of the genetic vaccines. For instance, if more death occurs in the vaccinated people, one should take into account that, among these people, there are many at-risk and elderly patients. An analysis should be conducted with awareness of this bias and should divide the cases in different classes of ages by estimating the percentage of at-risk people in the most affected population.

Repeated administrations (up to four or five and more) were not included in the seminal clinical trials of the vaccine makers, so the intensity and frequency of adverse events can now change in the face of an infection that has a current mortality comparable or even lower than that of flu [199]. No large human studies are available on updated mRNA products, which encode for two types of Spike proteins at the same time, regarding protection from the disease. In a recent report, immunogenicity of the bivalent vaccine was studied after 28 days, but the safety assessment stopped at day 7 [200]. Compared to other variants, the Omicron variant has at least three times more affinity for ACE2 (affinity is based on Spike protein interaction with its receptor) [24]. This may affect the function of ACE2 in a stronger manner after inoculation, when several Spike molecules of the Omicron type are translated and spread throughout the body. A paper in preprint analyzed, side by side, the adverse reactions to the old and bivalent vaccine among 76 healthcare workers and found more reactions and higher inability to work from the bivalent vaccine [201]. Other and more precise studies are needed for the bivalent and the former vaccines.

In this regard, a recent retrospective study, which was performed in one province in Italy, states that no increased risk of serious adverse events potentially caused by the vaccines could be observed in the reference population. The study claimed to have made observations for 18 months. However, from the tables presented, it seems that people vaccinated once, and especially those vaccinated twice, but not those vaccinated three times, have a higher risk of death from non-COVID-19-related causes and have double or triple the chances of having a heart infarction or a stroke, as compared to unvaccinated individuals. After the third dose, no relevant adverse events were noted. However, the follow-up of 18 months is valid only for the unvaccinated people because the vaccinated were followed only from the date of their first, second, or third dose. Indeed, the days of follow-up of the unvaccinated individuals are double, or more than double, those of the people with one, two, or three doses. It is unclear what makes only the triple-vaccinated people less susceptible to death and other accidents. There is a possibility, not discussed, that those who were less affected by the vaccines could have decided to receive the third doses more promptly. As also stated by the authors, further research in the coming years will be required to evaluate the long-term safety of the COVID-19 vaccines [202]. Other studies are needed. The risk of interference (also via the above-described mechanisms of TCR antagonism and immune imprinting) could be assessed, as this risk depends on the particular genetic background of each individual. The immune system is at risk when dealing with more than one epitope variant at once, and this risk involves outcomes that, at present, are not possible to forecast; among these outcomes, ADE can be envisaged as one the possible effects. “Anergy” of the T-cells involved in anti-viral immunity could result from continuous stimulation of the immune system. Although this is not proven, a recent paper published in *Science Immunology* shows how repeated boosts of mRNA-based vaccines, but not DNA-based vaccines, induced a class of antibodies (IgG4), which are anti-inflammatory and are endowed with poor effector functions (for instance, less antibody-dependent cytotoxicity, ADCC) [203]. IgG4 usually develops against allergens to protect the body against excessive immune responses. However, if this mechanism dampens the immune response to the virus in mRNA vaccine recipients, instead of inducing a protective response, then this process needs to be assessed. For the moment, we know that anti-Spike IgG4 antibodies were associated with more severe COVID-19 progression and poor prognosis in former studies [204,205]. Other conventional vaccines, which were studied by the authors in another paper [164], did not show induction of this IgG4 class, even after repeated inoculations [203]. Because the production of the right antibodies depends on T-cell help, tolerance in T-cells is an unwanted effect. With regard to induction of T cell anergy, which leads to tolerance, a recent paper demonstrated induction of both cellular and humoral tolerance after repetitive administration of vaccine boosters in a mouse model. The approach in this paper was to boost mice with repeated stimulations in a conventional manner, using a SARS-CoV-2 recombinant receptor-binding domain (RDB) protein. This resulted in a dramatic decrease of neutralizing anti-SARS-CoV-2 antibodies and impaired activation of CD4 and CD8 T-cells; T-cells showed acquisition of a phenotype, which promoted adaptive immune tolerance. This also means that the lost efficacy of the immune response might be independent of the vaccine type and may concern the negative effect of repeated stimulations toward a single antigenic determinant to narrow and focus the immune response [206].

At-risk people are not only elderly patients. Apart from cancer, which can affect both young and old patients, immune-mediated and autoimmune diseases such as diabetes, multiple sclerosis, psoriasis, and others can also develop in the young. Pediatric patients and young people with these chronic conditions can also be at risk of myocarditis development, as myocarditis cases are not rare in young people, as reported above. In the present review, we have reported frequencies of myocarditis cases of up to 1:300 (active survey) or 1:1000 (passive survey) in young and adolescent patients. When instrumental tests take place, these analyses revealed higher frequencies. In a recent paper, young patients with vaccine-induced myocarditis were followed for several months, and not all of the patients experienced resolved symptoms, although most patients responded to treatment. The authors demonstrated persistence of abnormal findings on cardiac MRI [207], and the elevation of other parameters that can be associated with poor outcomes. Myocarditis is a form of heart inflammation that can lead to future additional health issues in at-risk young patients with an already compromised chance of life. The scientific community needs to be aware and discuss whether the use of the current genetic COVID-19 vaccines, which was justified at the time of earlier deadly coronavirus variants, should still be encouraged at the time of Omicron variants. Another recent paper linked the formation of blood clots to vaccination with genetic vaccines in people aged 65 and over [208]. Thus, at this stage, the risk/benefit could be re-assessed also for elderly people. The development of more traditional vaccines based on antigens that are much less variable and that are not endowed with intrinsic toxic effects is highly desirable for protecting the elderly and at-risk people, including those with autoimmunity [209,210]. These vaccines should be able to induce IgA in addition to IgG to block transmission. A 2021 paper showed that IgA can be increased by COVID-19 mRNA vaccines, but only in people that had a previous SARS-CoV-2 infection and COVID-19 disease [211].

## Data Availability

Not applicable.

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
