# Peer review of "Safety of COVID-19 Vaccines in Patients with Autoimmune Diseases, in Patients with Cardiac Issues, and in the Healthy Population"

_pathogens, 2023, doi:10.3390/pathogens12020233_

Round 1

Reviewer 1 Report

The manuscript addresses the issue of COVID-19 mRNA vaccines-associated risks in patients with autoimmune disease and with cardiac issues. The topic is timely and relevant, and the study is enriched by a section dealing with potential pathogenic mechanisms, which is very important for the better evaluation of vaccines in the clinical context.

From a general point of view, the title and the text should better define what is meant by “cardiac issues” (actually, the authors discuss only cardiac inflammation). I’d recommend to include somewhere, e.g. in the Introduction, a couple of short paragraphs dealing with the well-known increased cardiovascular risk in people with autoimmune disease and on the other side the role of (auto)immunity in cardiac inflammation, to better provide the author with a homogeneous rationale for the study.

Abstract: please, avoid any reference to previous studies published anywhere.

In the introduction, please acknowledge that hundreds COVID-19 vaccines are in various stages of development worldwide, that they are based on many different technologies, including more conventional ones, and that the few information about efficacy suggests that risk reduction regarding symptomatic COVID-19 is more or less that same, even for more conventional vaccines (e.g. the Abdala and the Soberana from Cuba). This is important as a premise to help the reader understanding that the choice to use the COVID-19 mRNA vaccines is by no way based on scientific grounds.

Section 2.1 is entitled “COVID-19 vaccination among fragile individuals such as patients with autoimmunity” however there is no discussion at all about vaccination in autoimmune patients. Authors might want to consider the extensive evidence about e.g. multiple sclerosis, where COVID-19 vaccination has been associated with relapses. Concerning SLE (and autoimmunity in general) the issue of COVID-19 vaccine-induced immunodepression might be of interest (see e.g. https://pubmed.ncbi.nlm.nih.gov/36563528/). Another major autoimmune disease which should be considered at least is rheumatoid arthritis.

In Section 2.2, when discussing the different figures about post-vaccine myopericarditis frequency, please make clear that some studies consider only events recorded in hospital, thus excluding outpatients and subclinical cases (e.g. identified through instrumental/lab tests). In addition, is is important to point out that most studies usually exclude from the count events occurring in people with previous COVID-19 (since “it’s COVID-19”) and people with previous myopericarditis (since “it’s individual predisposition”). See e.g. https://www.nature.com/articles/s41467-022-31401-5.

Please move Table 1 after Section 2.2, in order to make the whole information more easily grabbed by readers. Please carefully revise the table for typos.

In Section 2.3, when commenting the paper by Ramirez et al., please make immediately clear that results show blood troponin increase in all the subjects, which is a direct marker of heart damage. In this regard, please also make clear that troponin always suggests myocardial damage, either ischemic or not ischemic (see e.g. https://www.sciencedirect.com/science/article/pii/S2542364917300997).

Beginning from Section 2.4, another major title should be chosen, e.g. 3. Mechanisms of COVID-19 mRNA vaccines-induced tissue/organ damage, or similar.

The issue of repeated boosters in non-responders is correctly mentioned, however it should also possibly be discussed in the light of this recent study: https://www.sciencedirect.com/science/article/pii/S2589004222017515

In Section 2.7, when discussing the potential of the Spike protein to trigger autoimmune responses, authors should consider also the hypothesis put forward in this paper: https://pubmed.ncbi.nlm.nih.gov/35298029/

Author Response

For reply see attached PDF file

Reviewer 2 Report

·         Authors delineates the safety and toxicological effects of COVID-19 vaccines in autoimmune patients.

·         The topic and theme of the manuscript are attractive and well-written.

·         Some typos and grammatical errors are embedded. I recommend authors to proofread and make changes if needed. Some

o   A typo in the title “AUTOMMUNE

o   Line # 108 “Myocarditis are”. Please insert cases after myocarditis.

o   Line # 363 “biopsies form”. From and not Form.

·         I recommend authors to insert the two reference (listed below) that are related to theme of manuscript:

o   Hajjo, R.; Sabbah, D.A.; Bardaweel, S.K.; Tropsha, A. Shedding the Light on Post-Vaccine Myocarditis and Pericarditis in COVID-19 and Non-COVID-19 Vaccine Recipients. Vaccines (Basel) 2021, 9, doi:10.3390/vaccines9101186.

o   Hajjo, R.; Sabbah, D.A.; Tropsha, A. Analyzing the Systems Biology Effects of COVID-19 mRNA Vaccines to Assess Their Safety and Putative Side Effects. Pathogens 2022, 11, doi:10.3390/pathogens11070743.

·         DECISION: ACCEPTED PROVIDED TYPOS AND ERRORS ARE CORRECTED

Author Response

For the reply see response to reviewer 2.

Round 2

Reviewer 1 Report

The authors satisfactorily replied to the points raised in the initial round of revision.